# ILC3 Function as a Double-Edged Sword in EV71 Infection

**DOI:** 10.3390/v17020184

**Published:** 2025-01-27

**Authors:** Chang Zhang, Linlin Bao, Feifei Qi, Qi Lv, Fengdi Li, Chuan Qin

**Affiliations:** 1Beijing Key Laboratory for Animal Models of Emerging and Remerging Infectious Diseases, Institute of Laboratory Animal Science, Chinese Academy of Medical Sciences, Beijing 100021, China; chang_z@yeah.net (C.Z.);; 2NHC Key Laboratory of Human Disease Comparative Medicine, Comparative Medicine Center, Peking Union Medical College, Beijing 100021, China

**Keywords:** HFMD, EV71, ILC3, IL-17, IL-22

## Abstract

Enterovirus 71 (EV71) is a common pathogen responsible for hand, foot, and mouth disease (HFMD), leading to severe neurological complications and even death. However, the mechanisms underlying severe EV71-induced disease remain unclear, and no effective specific treatments are available. In this study, we successfully infected mice of different ages using a mouse-adapted EV71 strain, resulting in disease and mortality. We compared immune system responses between infected and uninfected mice of different ages to identify key pathogenic targets during EV71 infection. Our findings revealed that the level of Group 3 Innate Lymphoid Cells (ILC3s) in mice negatively correlated with the severity of disease induced by EV71 infection. We conducted anti-ILC3 cytokine injections and cytokine neutralizing antibody experiments on 14-day-old EV71-infected mice. The results showed that the cytokine IL-17 secreted by ILC3 cells had a mild protective effect, while IL-22 promoted inflammatory responses. Our research demonstrates that ILC3 cells play a dual role in EV71 infection. These findings not only clarify key immune factors in the progression of EV71-induced disease but also provide a promising approach for the early diagnosis and treatment of severe EV71 infections.

## 1. Introduction

Enterovirus 71 (EV71) can cause hand, foot, and mouth disease (HFMD) in infants and children. In severe cases, it can progress to serious neurological symptoms and even be life-threatening. Severe HFMD cases often involve persistent neurological complications, delayed neurodevelopment, and cognitive decline. EV71 accounts for approximately 90% of these severe cases and is especially prevalent in economically underdeveloped regions with poor public health management in the Asia-Pacific region. HFMD outbreaks show a cyclical pattern, with major epidemics occurring every 2–3 years [1]. The largest recent EV71 outbreak occurred in 2008 in Fuyang, Anhui Province, China, with 490,000 reported infections and 126 deaths [2]. During a previous EV71 epidemic in Bulgaria (1975), HFMD primarily manifested with central nervous system symptoms, resulting in hundreds of reported cases, including 545 cases (77.3%) of aseptic meningitis and 149 cases (21.1%) of acute flaccid paralysis. There were 44 deaths (6.2%) during this outbreak, mostly due to bulbar damage [3]. Notably, there are currently no effective antiviral drugs to treat EV71 or HFMD caused by other viruses [4]. Supportive symptomatic treatment and intravenous antibody administration are the only available options for severe cases, though these antibodies target the virus itself and appear to have limited efficacy in mitigating the inflammatory response [5]. Vaccines against EV71 have shown preventive effects in clinical trials but are ineffective in already infected individuals [6,7]. HFMD progresses rapidly from mild to severe, and there are no effective methods or indicators for early identification of severe cases [8,9].

It is widely believed that innate immunity plays a crucial role in the antiviral response during EV71 infection [10]. The gut, as the second largest immune organ, has significant immune functions and is the first organ infected during natural EV71 infection, making it a key target for early-stage viral replication [11,12]. This study analyzed the proportions of intestinal immune cells in EV71-infected mice of different ages to identify key factors influencing EV71 susceptibility and disease progression. By analyzing the proportion of ILC3 cells in EV71-infected mice of different ages, we found that changes in ILC3 levels were closely related to the severity of disease induced by EV71 infection. We hypothesize that ILC3 cells and their cytokines, IL-17 and IL-22, may also play a role in EV71 infection. This study validated this hypothesis in EV71-infected mice.

## 2. Materials and Methods

### 2.1. Cells and Virus

RD cells (human rhabdomyosarcoma cells) were cultured according to the instructions provided by the American Type Culture Collection. The EV71 strain GD 10–12 (GenBank accession number KJ004559) was isolated from a patient in Guangdong Province in 2010. The GD10–12 virus, amplified in RD cells, was injected intraperitoneally into SPF-grade 10-day-old BALB/c mice at a dose of 3 × 10^8^ pfu per gram of body weight. On the third day post-infection, the mice were euthanized, and the skeletal muscle tissue from the hind limbs was collected. The virus was separated by grinding the tissue and then amplified in RD cells. The amplified virus was used to reinfect mice, and this process was repeated for 10 passages to obtain the GD 10–12 mouse-adapted strain.

### 2.2. Animal Infection

SPF-grade BALB/c mice aged 7–21 days, with 5 to 7 pups per litter, were naturally nursed by their mothers. The experimental animals were purchased from Beijing Vital River Laboratory Animal Technology Co., Ltd. (Beijing, China) [SCXK (Jing) 2007-0001]. The mice were housed in RVC-II type individually ventilated cages (IVCs) from TENCNIPLAST in an SPF-grade animal facility. Mice were infected intraperitoneally with EV71 virus at a dose of 3 × 10^8^ pfu per gram of body weight. After infection, the animals were kept in IVCs with a filtration system. All infectious procedures were conducted in a biosafety level 2 laboratory. The animal experiments were reviewed and approved by the Ethics Committee of the Institute of Laboratory Animal Science, Chinese Academy of Medical Sciences & Peking Union Medical College (BLL17011). Post-infection, the survival rate, body weight, and clinical scores of the mice were recorded daily for 15 days. The clinical scoring criteria for the infected mice were as follows: 0—healthy, 1—ruffled fur, 2—hind limb weakness, 3—paralysis of one hind limb, 4—paralysis of both hind limbs, 5—death.

### 2.3. Virus RNA Extraction, Quantitative Reverse Transcription PCR, and Whole Genome Sequencing

Total RNA was extracted from tissues using the RNeasy Mini Kit (Qiagen, Hilden, Germany) according to the manufacturer’s instructions. The sequences of the EV71 primers were as follows: forward primer (5′-AGATAGAGTGGCAGATGTAAT-3′) and reverse primer (5′-TTGATGATGCTCCAATCTCAG-3′), with the TaqMan probe sequence (FAM-CTCTACCAGCACACACAGGCCAGAAC-BHQ1). Polymerase chain reaction (PCR) was performed in a 25 µL reaction volume containing the master mix components from the QuantiTect Probe RT-PCR Kit (Qiagen). PCR amplification began with an incubation at 25 °C for 10 min, followed by 50 °C for 50 min, and then 35 cycles of 30 s at 94 °C, 45 s at 54 °C, and 60 s at 68 °C. All reactions were performed in triplicate. Virus genome sequencing was conducted using the Sanger method, with the primers for the full genome sequencing of EV71 listed in Table A1.

### 2.4. Tissue Virus Titer Determination

On the third day post-infection, mice were euthanized and the skeletal muscle from the hind limbs was collected and placed in EP tubes containing virus maintenance solution. The tissue was ground and then sonicated until the solution became clear. After centrifugation, the supernatant was collected, discarding the pellet. RD cell suspensions were added to a 6-well plate and grown for 8–12 h until they formed a monolayer. The tissue homogenate was serially diluted 10-fold with maintenance solution. The virus dilutions were added to the 6-well plates containing monolayer RD cells, with 1 mL per well, and incubated at 37 °C with 5% CO_2_ for 60 min. A 1% low-melting-point agarose solution was autoclaved at 121 °C for 15 min. The virus solution was aspirated, and 1% agarose, preheated to 38 °C, was added. The plates were left at room temperature to allow the agarose to solidify. The 6-well plates were then incubated upside down at 37 °C with 5% CO_2_ for 3 days. Plaques were counted under an optical microscope (LEICA DM2000, Mannheim, Germany) to calculate the virus titer.

### 2.5. Histopathological Examination

Necropsy of the animals was performed according to established protocols. At specific time points, mice were euthanized, and their tissues were dissected and placed in 10% formalin overnight for fixation. After fixation, the tissues were embedded in paraffin, sectioned into 5-micron thick slices, and stained with hematoxylin and eosin (HE). The stained sections were then observed under an optical microscope (LEICA DM2000, Mannheim, Germany).

### 2.6. Flow Cytometry Analysis of Small Intestinal Tissue

At specific time points, three mice were euthanized, and their small intestines were collected for flow cytometry analysis. Preparation of single-cell suspensions involved cutting the intestinal tissue into fragments smaller than 2 mm and placing them in a 50 mL centrifuge tube containing EDTA (Gibco, Waltham, MA, USA) up to 45 mL. The mixture was vortexed for 30 s to thoroughly disperse the tissue fragments and then incubated in a water bath at 37 °C with 160 rpm shaking for 15 min. After allowing the tissue fragments to settle for 1 min, the supernatant was aspirated with a syringe, leaving 5 mL of liquid and tissue fragments in the centrifuge tube. RPMI-1640 medium (Gibco) containing 20% fetal bovine serum, pre-warmed to 37 °C, was added to the tube up to 20 mL, followed by the addition of 1 mL collagenase (Sigma, Darmstadt, Germany) and 50 µL DNase (Sigma). The mixture was vortexed for 30 s to disperse the tissue fragments thoroughly and then incubated in a water bath at 37 °C with 160 rpm shaking for 60 min. The suspension was then centrifuged at 400× *g* at 4 °C for 5 min, and the supernatant was discarded. The pellet was resuspended in an appropriate volume of HBSS (Gibco) for further use.

For cell staining, the cells were centrifuged again at 400× *g* at 4 °C for 5 min and resuspended in 100 µL Stain Buffer (BD Pharmingen, San Diego, CA, USA). Flow cytometry antibodies (BD Pharmingen) were added for staining, and the cells were incubated at 4 °C in the dark for 20 min. ILC3 cells were identified as lin-, RORγt+, NKp46+.

### 2.7. Post-Infection Intervention

Twenty-four hours after infection, a variety of intervention methods were administered to the mice. Fourteen-day-old mice were divided into groups for daily interventions. The control group mice received daily injections of 200 µL phosphate-buffered saline (PBS, Gibco). Four intervention groups received daily injections of either IL-17 (Abcam, ab281805, Cambridge, UK), IL-22 (Abcam, ab281809), anti-IL-17 neutralizing antibody (Thermo Fisher, PA547287, Waltham, MA, USA), or anti-IL-22 neutralizing antibody (Thermo Fisher, PA547782) at 200 µL per mouse. The dosage of cytokines or antibodies for intervention was 20 µg per mouse per day, administered for five consecutive days. The survival rate, body weight, and clinical scores of the infected mice were recorded daily for 10 days post-infection.

### 2.8. Detection of Cytokine mRNA in Tissues

On the third day post-infection, fresh tissue samples (10 mg each) were collected from animals, and total RNA was extracted from each tissue using the RNeasy Mini Kit (Qiagen, Hilden, Germany) according to the manufacturer’s instructions. The primers used for detecting the relative expression levels of cytokine mRNA were as follows: IL-17A forward primer (5′-GACTACCTCAACCGTTCCACGTC-3′) and reverse primer (5′-TCTATCAGGGTCTTCATTGCG-3′), IL-22 forward primer (5′-TTGTGCGATCTCTGATGGCT-3′) and reverse primer (5′-CCAGCATAAAGGTGCGGTTG-3′). Polymerase chain reaction (PCR) was performed in a 20 µL reaction volume containing the master mix components from the QuantiTect Probe RT-PCR Kit (Qiagen). The PCR amplification started with incubation at 25 °C for 10 min, followed by 50 °C for 50 min, and then 35 cycles of 5 s at 94 °C, 30 s at 60 °C, and 15 s at 95 °C. All reactions were performed in triplicate. The RNA expression levels were calculated using the RO value = 2^ΔΔCT^ method.

### 2.9. Detection of Cytokine Protein in Tissues

On the third day post-infection, fresh tissue samples (10 mg each) were collected from animals and homogenized. The cytokine levels were detected using the Mouse IL-17A ELISA Kit (ab199081) and the IL-22 Mouse ELISA Kit (Thermo Fisher) following the manufacturer’s instructions through enzyme-linked immunosorbent assay (ELISA).

### 2.10. Statistical Methods

Differences in mortality, body weight, and subsequent paralysis sequelae of mice were evaluated by mean ± SE values. *p*-values were evaluated by log-rank test (survival rates), Wilcoxon test (clinical scores), and Mann–Whitney U test (relative change in ILC3s, IL-22, and IL-17). *p*-values < 0.05 were considered statistically significant.

## 3. Results

### 3.1. The Disease Progression of EV71 Virus Infection Varies in Mice of Different Ages

This study employed 10 clinical isolates of EV71 and one mouse-adapted strain to infect suckling mice, with subsequent passages in mice to adapt the virus (Appendix B). A highly virulent strain, GD10–12, was selected and used to infect mice older than 14 days, causing disease. Mice aged 15–18 days and 19–21 days exhibited different disease progression patterns after infection with GD10–12 (Figure 1).

Infection of 15–18-day-old BALB/c mice with GD10–12 resulted in mortality, with a 100% mortality rate within 7 days post-infection (dpi) (Figure 1a), while no mortality was observed in 19–21-day-old mice. Weight loss (Figure 1b), hind limb paralysis (Figure 1c,d), and ocular inflammation (Figure 1e) were observed in mice of all age groups. For 15–18-day-old mice infected with GD10–12, as the age increased, the time to reach 100% mortality gradually extended, and the onset of clinical symptoms also delayed. Although 19–21-day-old mice infected with GD10–12 did not experience mortality, they all developed ocular lesions. The onset of ocular lesions was delayed with increasing age, and the proportion of mice with ocular lesions decreased gradually with age.

On the third dpi, viral titers were detected in the brain, spinal cord, skeletal muscle, duodenum, jejunum, ileum, rectum, lungs, thymus, spleen, liver, heart, and kidney tissues (Figure 1f). On the third dpi, virus was detected in the tissues of mice of all ages. The viral titers in the tissues of 15–18-day-old mice were generally similar but gradually decreased with increasing age, with the lowest titers observed in 18-day-old mice. Among all tissues, the highest viral titers were found in skeletal muscle, and within the intestinal segments, the ileum had the highest viral titer. These findings suggest that skeletal muscle and the ileum are the sites of most active viral replication in mice. As the ileum is a highly active site of immune activity in the body, its various immune components may play a crucial role in the disease progression following infection.

To analyze the histopathological changes in mice after EV71 infection, histopathological analysis was conducted on 18-day-old mice, which were the oldest age group susceptible to lethal outcomes from GD10–12 infection (Figure 1g). The results showed necrosis in the hind limb skeletal muscle on the fifth dpi, along with vacuolar degeneration in the epithelial cells of the jejunum. No definite histopathological changes were observed in other tissues (ileum, colon, eyes, brain, spinal cord).

In this study, we screened the EV71 mouse-sensitive strain GD10–12 and obtained a mouse-adapted mutant strain based on it. We found that infection of 15–18-day-old mice resulted in 100% mortality, while infection of 19–21-day-old mice caused disease without mortality. By comparing the changes in mice of different ages after infection, we found that the hind limb skeletal muscle tissue exhibited the strongest disease response.

### 3.2. The Number of ILC3s in the Small Intestine of Mice Significantly Decreased After EV71 Infection

The number of ILC3s in the small intestine of mice significantly decreased after EV71 infection, and the proportion of this decrease was negatively correlated with the age of the mice. To further observe the impact of the internal environment of mice on EV71 infection, this study focused on analyzing the changes in the jejunum of four representative groups of mice of different ages (7, 14, 18, and 21 days old) after EV71 infection. The survival rate, weight changes, and clinical manifestations varied among mice of different ages after EV71 infection (Figure 2).

Infection with GD10–12 resulted in mortality in 7-, 14-, and 18-day-old mice, with a 100% mortality rate reached within 3, 4, and 7 days post-infection, respectively. However, no mortality was observed in 21-day-old mice after infection (Figure 2a). Weight loss was observed in 7-, 14-, and 18-day-old mice, but not in 21-day-old mice (Figure 2b). All mice exhibited piloerection and ocular lesions, while only 7-, 14-, and 18-day-old mice developed hind limb paralysis (Figure 2c). As the age of the mice increased among the 7-, 14-, and 18-day-old groups, the time to reach 100% mortality gradually extended, and the onset of clinical symptoms also delayed.

Pathological examination of the small intestine tissues revealed varying degrees of tissue lesions in mice of different ages (Figure 2d). After infection with GD10–12, both 7-day-old and 14-day-old mice exhibited vacuolar degeneration in the small intestinal epithelium, with a more extensive range observed in 7-day-old mice. However, no significant vacuolar degeneration was observed in the small intestinal epithelium of 18-day-old and 21-day-old mice after infection. This result indicates that the small intestine of mice of different ages responds differently to viral infection.

Flow cytometry results showed that compared to uninfected mice (control group), all age groups of mice experienced a decrease in the number of ILC3 cells after infection (Figure 2e). As the age of the mice increased, the number of ILC3s in the small intestine increased (Figure 2f), and the disease response of the mice to EV71 infection became less severe. Simultaneously, the loss rate of ILC3s after infection decreased (Figure 2g). This may be due to the enhanced resistance of mice to the pathological damage caused by EV71 as they mature. These results suggest that ILC3s are involved in the pathological process of the disease and may affect the susceptibility of mice of different ages to the virus and the speed of disease progression.

### 3.3. IL-17 Is Negatively Correlated with the Severity of Disease in Mice, While IL-22 Is Positively Correlated with the Severity of Disease

Flow cytometry analysis revealed that ILC3s were significantly decreased in all age groups of mice compared to the control group after EV71 infection, indicating that ILC3s play a role in the disease pathology and are likely key factors influencing the susceptibility to EV71 and disease progression in mice of different ages. Cytokine secretion is an important pathway for immune cells to participate in the immune process of the body. Therefore, we hypothesize that ILC3s may affect the disease progression of EV71 infection through their main cytokine products. To verify this hypothesis, we subsequently detected the mRNA transcription and protein expression of IL-17 and IL-22, the main cytokine products of ILC3s, in the small intestine of mice of different ages on the third day after EV71 infection. Additionally, we conducted an intervention experiment in 14-day-old mice by administering IL-17/22 and anti-IL-17/22 neutralizing antibodies to EV71-infected mice.

The results showed that IL-17 was negatively correlated with the severity of disease in mice, while IL-22 was positively correlated with the severity of disease. In the intervention experiment, IL-17 cytokine and anti-IL-22 neutralizing antibody exhibited protective effects on mice (Figure 3).

There were differences in the mRNA transcription and protein expression levels of IL-17 and IL-22 between mice that died (7, 14, and 18 days old) and those that survived (21 days old) after infection. The expression levels of IL-17 and IL-22 may be important factors affecting the survival rate of mice after infection (Figure 3a–c). In the post-infection intervention experiment, IL-17 partially reduced the mortality rate of mice, with a survival rate of 60% (12/20). Anti-IL-22 neutralizing antibody significantly reduced the mortality rate of infected mice, with a survival rate of 80% (16/20), which was significantly higher than that of mice treated with PBS (*p* < 0.05, log-rank test). In contrast, all mice in the PBS treatment group died (0%; 0/10) (Figure 3d). All infected mice experienced weight loss (Figure 3e), piloerection, hind limb paralysis (Figure 3f), and were accompanied by kyphosis, lethargy, ataxia, and reduced food intake. Mice treated with IL-17 did not show obvious signs of inflammation on the fourth day after infection, and mice treated with anti-IL-22 antibody did not show obvious signs of inflammation on the third day after infection, while mice in the PBS injection group began to develop hind limb paralysis on the second day after infection (Figure 3f). The results indicated that both IL-17 and anti-IL-22 antibodies could reduce the mortality rate of mice and had protective effects.

Although there was no significant difference in tissue viral titers between the intervention group and the untreated infected mice (Figure 3g), the range of skeletal muscle necrosis was smaller in the intervention group than in the untreated group (Figure 3h). Moreover, the onset of skeletal muscle necrosis in the IL-22 intervention group (on the sixth day after infection, or the fifth day after intervention) was delayed compared to the untreated group (on the third day after infection).

## 4. Discussion

Currently, 10-day-old mice are commonly used in EV71-related research. Although these young mice are more susceptible to successful EV71 infection, the disease progression is relatively short, with most mice dying within 5 days. The immature immune organs and tissues of these mice, coupled with the short disease course, limit the information that can be obtained from studying the interaction between EV71 virus and the host immune system. In this study, we successfully infected older mice with the EV71 GD10–12 mouse-adapted strain, providing a valuable research tool for EV71-related studies. Infection of 15–18-day-old mice with the GD10–12 strain results in death, while infection of 19–21-day-old mice causes disease without lethality. This makes it possible to study the interaction between postnatal development and EV71, offering a more convenient and observable tool for future EV71-related disease research and providing greater flexibility for other studies.

In recent years, the role of ILC3s in intestinal immunity has gradually attracted the attention of researchers. ILC3s are mainly located in the intestine and other mucosal tissues, and they regulate inflammatory responses, tissue repair, and maintain intestinal barrier function and balance of the intestinal microbiota by secreting cytokines such as IL-17 and IL-22. The production of these two cytokines is regulated by the cytokine network in the microenvironment. IL-23 and IL-1β have been shown to enhance the ability of ILC3s to produce IL-17 and IL-22 [7]. In this study, we found that the number of ILC3s in the small intestine significantly decreased in mice of different ages after EV71 infection. As the age of the mice increased, the proportion of ILC3s decreased gradually after infection, and the severity of the disease in the mice gradually decreased. This difference may be related to the maturity of the mouse immune system, indicating that innate immunity, especially ILC3s, plays an important role in the susceptibility of mice to EV71 and the progression of the disease after infection.

Although the protective role of IL-17 in viral immunity and its mechanism are not yet clear, there is evidence that ILC3s secrete IL-17 when encountering bacterial or fungal infections in the intestine, especially Gram-negative bacteria and some intestinal pathogens such as Salmonella and enteropathogenic *Escherichia coli*. IL-23 and IL-1β are key factors that promote ILC3s to secrete IL-17. IL-23, secreted by dendritic cells and macrophages, enhances IL-17 production by binding to the IL-23 receptor on ILC3s [8]. IL-1β further enhances IL-17 secretion by activating the NF-κB signaling pathway. This production of IL-17 mainly acts in the early stages of infection, promoting the recruitment and activation of neutrophils, enhancing the local inflammatory response, and helping to clear pathogens. This is consistent with our observation that IL-17 exhibits a clear protective effect in the early stages of EV71 infection in mice, reducing the severity of the disease by inhibiting the inflammatory response and alleviating tissue damage [9]. The expression level of IL-17 mRNA was negatively correlated with the severity of the disease in mice of different ages. In the intervention experiment, IL-17 injection delayed the onset of clinical symptoms in mice and reduced their mortality to a certain extent, alleviating the inflammatory necrosis response in skeletal muscle. This result may be due to the protective effect of IL-17 on the small intestinal mucosa, an early site of virus infection and proliferation, in the early stages of mouse infection [10].

Current reports on the role of IL-22 mainly focus on its effects on epithelial cells and fibroblasts in various organs. ILC3s mainly secrete IL-22 to maintain the integrity of the intestinal epithelial barrier [11]. The secretion of IL-22 is also regulated by IL-23, but its secretion is more influenced by environmental factors such as the intestinal microbiota and local damage signals [12]. When the intestinal epithelium is damaged or invaded by pathogens, ILC3s secrete IL-22 to promote the proliferation and regeneration of epithelial cells and help restore barrier function by inducing the production of antimicrobial peptides [13]. For example, IL-22 maintains intestinal barrier function by promoting the proliferation of epithelial cells and the secretion of antimicrobial peptides. IL-22 can also trigger the expression of chemokines in synergy with other cytokines, recruiting and activating immune cells to resist pathogen infection. In addition, IL-22 can act on epithelial stem cells to promote their protection and proliferation, thereby contributing to the repair of epithelial damage. The secretion of IL-22 is usually activated in the early stages of infection or tissue damage and continues throughout the recovery process [14].

The dynamic changes in the secretion of IL-17 and IL-22 by ILC3s are closely related to the progression of the disease. In the early stages of the disease, especially during acute infection, ILC3s tend to secrete higher levels of IL-17 to rapidly mobilize neutrophils and initiate an inflammatory response [15]. During this period, the level of IL-17 rapidly rises in response to the presence of pathogens.

As the disease progresses, especially in chronic inflammation or the recovery phase, the function of ILC3s gradually shifts to secreting IL-22 [16]. The secretion of IL-22 is more prominent in the late stages of the disease, as the body needs to repair damaged tissues and maintain the integrity of the epithelial barrier. Studies have also found that in the context of chronic inflammation or inflammatory bowel diseases (such as Crohn’s disease and ulcerative colitis), ILC3s may continuously secrete IL-22 to try to balance ongoing tissue damage and repair needs.

However, there is currently a lack of research and reports on the effects of IL-22 on non-epithelial cells. Unlike previous studies, our research found that during EV71 infection, IL-22 promotes the inflammatory response in mouse skeletal muscle, exacerbating inflammatory necrosis and thus increasing the severity of the disease. The expression level of IL-22 mRNA was positively correlated with the severity of the disease [17]. In the intervention experiment, injection of IL-22 neutralizing antibody slightly delayed the onset of clinical symptoms in mice but significantly reduced their mortality and markedly alleviated the inflammatory necrosis response in skeletal muscle. This suggests that IL-22 plays a role in promoting destructive inflammatory responses during EV71 infection in mice and is a damage-promoting factor in the disease process. This is likely the reason why anti-IL-22 antibodies have a protective effect on EV71-infected mice.

Although this study found that IL-22 has a pro-inflammatory effect in EV71 infection, its specific mechanism of action still requires further investigation, particularly its relationship with the severity of HFMD [18]. Of course, some studies have shown that IL-22 may have a protective effect rather than a pro-inflammatory effect in certain viral infections. For example, studies have found that IL-22 can promote the repair of epithelial cells and reduce lung damage during influenza virus infection. This result differs from the pro-inflammatory effect of IL-22 in EV71 infection observed in this study, suggesting that the role of IL-22 may depend on the specific infection environment and virus type.

The above results illustrate that ILC3 cells play an important role in the process of EV71 infection through the secretion of the cytokines IL-17 and IL-22, and their effects are dual-faced, with both anti-inflammatory and pro-inflammatory aspects. Therefore, the regulation of ILC3 function is likely a key factor influencing the progression of EV71 infection. Future studies are needed to investigate their function in other organs, especially the nervous system, to comprehensively understand their mechanism of action in EV71 infection.

This study also has some limitations. Firstly, our research was conducted only in a mouse model, and whether the results apply to humans still needs further verification. Secondly, the role of ILC3 cells may differ in different tissues, and further research is needed to investigate their function in other organs. In addition, the specific mechanism of action of IL-22 and its relationship with the severity of HFMD still require further investigation.

## 5. Conclusions

This study represents the first systematic analysis of the role of ILC3 cells and their secreted cytokines in EV71 infection, providing a new perspective for a deeper understanding of the pathogenic mechanisms of EV71. Furthermore, our research findings indicate that by regulating the expression or function of IL-22, tissue damage caused by EV71 infection can be effectively inhibited. In summary, this study uncovers the dual roles of ILC3 cells in EV71 infection, particularly the distinct roles of IL-17 and IL-22 in disease progression. Our research results offer new directions and potential therapeutic targets for the early diagnosis and treatment of EV71 infection, holding significant theoretical and practical value.

## Figures and Tables

**Figure 1 viruses-17-00184-f001:**
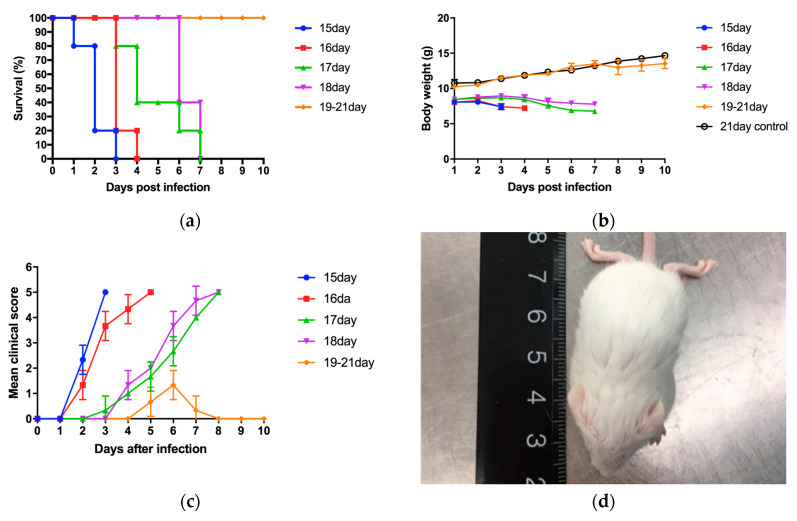
Different disease progression patterns after infection with GD10–12 of mice aged 15–18 days. (**a**) Survival rate of 15–21 day-old mice after infection. (**b**) Weight changes in 15–21 day-old mice after infection. The control group consists of uninfected mice. (**c**) Changes in clinical symptom scores of 15–21 day-old mice after infection. (**d**) Piloerection and hind limb paralysis observed in mice after infection. (**e**) Ocular inflammation and squinting observed in mice after infection. (**f**) Virus titers in different tissues on day 3 post-infection (dpi) in mice. (**g**) Histopathological changes in mouse tissues after EV71 infection (HE × 200). 1. Necrosis in skeletal muscle (black triangle). 2. Normal skeletal muscle in mice. 3. Vacuolar degeneration in jejunal epithelial cells (black triangle). 4. Normal jejunum in mice.

**Figure 2 viruses-17-00184-f002:**
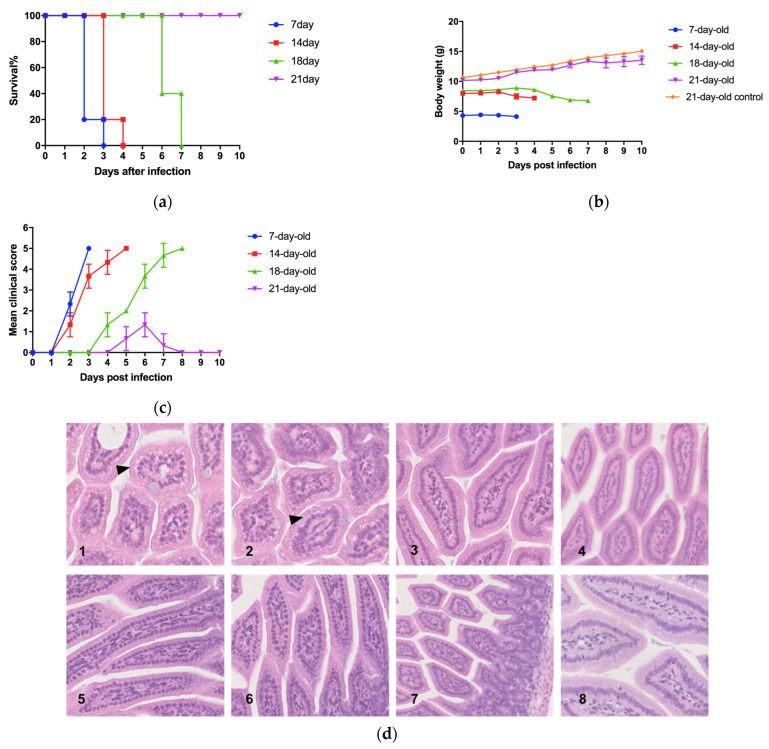
(**a**) Survival rate of 7–21 day-old mice after infection. (**b**) Weight changes in 7–21 day-old mice after infection. The control group consists of uninfected mice. (**c**) Changes in clinical symptom scores of 7–21-day-old mice after infection. (**d**) After challenging mice of different ages and obtaining small intestine tissues at 3 dpi, the images show typical tissue photographs (HE × 200) for each group, with black triangles indicating typical pathological changes. No significant pathological changes were observed in the tissues of the control group mice. 1. Seven-day-old mouse, challenge group, 2. Fourteen-day-old mouse, challenge group. 3. Eighteen-day-old mouse, challenge group. 4. Twenty-one-day-old mouse, challenge group. 5. Seven-day-old mouse, control group. Fourteen-day-old mouse, control group, 7. Eighteen-day-old mouse, control group. 8. Twenty-one -day-old mouse, control group. (**e**) Flow cytometry count results of ILC3 cells in the small intestine of mice of different ages before infection and on the third day post-infection. The control group consists of uninfected mice. (**f**) The ratio of ILC3 cell counts in the small intestine of mice of different ages on the third day post-infection to the ILC3 cell counts before infection, as determined by flow cytometry. The control group consists of uninfected mice. (**g**) The ratio of ILC3 cell counts in the small intestine of mice of different ages on the third day post-infection to the ILC3 cell counts in uninfected mice (control group), as measured by flow cytometry.

**Figure 3 viruses-17-00184-f003:**
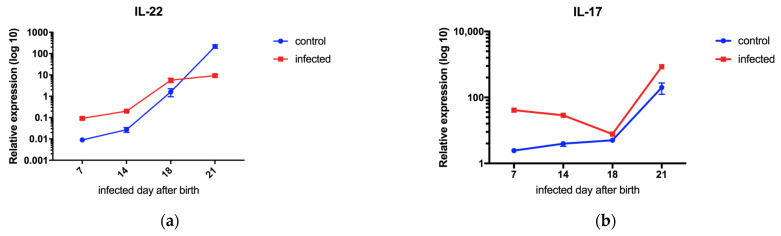
(**a**) Relative expression levels of IL-22 mRNA in mice of different ages with or without viral infection. The data were collected on the third day post-infection. (**b**) Relative expression levels of IL-17 mRNA in mice of different ages with or without viral infection. The data were collected on the third day post-infection. (**c**) Tissue levels of IL-22 and IL-17 in mice of different ages with or without viral infection. The data were collected on the third day post-infection. (**d**) Survival rates of mice after intervention with PBS, IL-22AB, or IL-17 following infection. (**e**) Changes in body weight of mice after intervention with PBS, IL-22AB, or IL-17 following infection. (**f**) Changes in clinical scores of mice after intervention with PBS, IL-22AB, or IL-17 following infection. (**g**) Tissue viral titers in mice after intervention with PBS, IL-22AB, or IL-17 following infection. The data were collected on the third day post-infection. (**h**) Skeletal muscle tissue of mice: 1. Mice on day 7 after IL-17 intervention following infection. 2. Mice on day 7 after IL-22AB intervention following infection. 3. Mice on day 10 after IL-17 intervention following infection. 4. Mice on day 10 after IL-22AB intervention following infection. 5. Uninfected mice. 6. Mice with PBS intervention. (Differences in mortality, body weight, and subsequent paralysis sequelae of mice were evaluated by mean ± SE values. *p*-values were evaluated by log-rank test (survival rates), Wilcoxon test (clinical scores), and Mann–Whitney U test (relative change in ILC3s, IL-22, and IL-17). *p*-values < 0.05 were considered statistically significant. * indicates *p* < 0.05).

## Data Availability

Data are contained within the article.

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
