# Peer review of "ILC3 Function as a Double-Edged Sword in EV71 Infection"

_viruses, 2025, doi:10.3390/v17020184_

Round 1
Reviewer 1 Report
Comments and Suggestions for Authors
1. The figure legend in figure 1 (1a, 1b, 1c) should reflect that the group currently labeled 21day is actually comprised of 19-21 day-old mice. The last group is currently only labeled 21day.
2. In figure 3c, the authors should adjust the Y-axis so that the data does not cutoff the error bars (i.e. 18day - 7 days after infection).
3. Figure 1 needs a title that summarizes all of the data that is included.
4. The explanation for Figure 1 legends (1a, 1b, and 1c) indicates only 15-18 day-old mice even though data from day 19-21day mice is included. For example, 1a says "Survival rate of 15-18 day-old mice after infection" but does not reference the 19-21day mice.
5. An description of how long the mice were observed for mortality is needed in the materials and methods section. The survival curves show data until day 10 post-infection but it would be good to know of the mice were observed longer than 10 days after infection.
6. Do the authors have normal (uninfected) controls for the weight loss data for both Figures 1b and 2b? The authors cite lack of weight loss in 21-day-old mice in Figure 2b but the weight gain appears to be less than would be expected in uninfected mice. Normal (uninfected) controls would provide a helpful baseline in evaluating this data.
7. Figure 3a and 3b the authors should correct the Y-axis titles to Relative Expression rather than "Relative Express". The figure legends for 3a and 3b should also be clarified to detection of mRNA for IL-22 and IL-17 rather than just RNA.
8. It is not clear from the materials and methods or from the data presented in Figure 3 when the samples were collected for mRNA and ELISA data of IL-22 and IL-17. This should be clarified to indicate day of tissue collection, preferably in both locations.
9. For the post-infection intervention study, it is not clear when the first treatments began. This should be clarified in section 2.7.
Reviewer 2 Report
Comments and Suggestions for Authors
This study investigates Enterovirus 71 (EV71), a primary cause of hand, foot, and mouth disease (HFMD) that can lead to severe complications. The authors infected mice of varying ages with a mouse-adapted strain of EV71 to evaluate immune responses. The study revealed a negative correlation between the presence of Group 3 Innate Lymphoid Cells (ILC3) and the severity of the disease. Furthermore, cytokine injection experiments demonstrated that IL-17 derived from ILC3 cells offered mild protection, whereas IL-22 aggravated inflammation. These findings highlight the dual role of ILC3 cells in EV71 infection and suggest potential pathways for the early diagnosis and treatment of severe cases. However, critical data are missing.
Major concern:
The key conclusion indicates a negative correlation between the presence of Group 3 Innate Lymphoid Cells (ILC3) and the severity of the disease. However, no relevant data is provided to support this conclusion.
Minor Points:
(1) Lines 239-241 indicate that Figures 2E-G are missing from Figure 2.
(2) Lines 258-259, there is no flow cytometry analysis presented in the whole manuscript.
(3) Statistical information is absent from Figure 3.
Comments on the Quality of English Language
The quality of the writing requires improvement.
Round 2
Reviewer 2 Report
Comments and Suggestions for Authors
I have reviewed the revised version, and the authors have adequately addressed all my concerns.